# Effectiveness of Colorectal Cancer Screening Promotion Using E-Media Decision Aids: A Systematic Review and Meta-Analysis

**DOI:** 10.3390/ijerph18158190

**Published:** 2021-08-02

**Authors:** Nur Suhada Ramli, Mohd Rizal Abdul Manaf, Mohd Rohaizat Hassan, Muhamad Izwan Ismail, Azmawati Mohammed Nawi

**Affiliations:** 1Department of Community Health, Faculty of Medicine, Universiti Kebangsaan Malaysia, Jalan Yaakob Latif, Cheras, Kuala Lumpur 56000, Malaysia; nursuhadaramli99@gmail.com (N.S.R.); mrizal@ppukm.ukm.edu.my (M.R.A.M.); rohaizat@ppukm.ukm.edu.my (M.R.H.); 2Ministry of Health, Malaysia, Federal Government Administrative Centre, Putrajaya 62514, Malaysia; 3Department of Surgery, Hospital Sultanah Aminah, Jalan Persiaran Abu Bakar Sultan, Johor Bahru 80100, Malaysia; izwanismail99@gmail.com

**Keywords:** colorectal cancer screening, effectiveness, electronic media, decision aids, systematic review, meta-analysis

## Abstract

Colorectal cancer (CRC)-screening reduces mortality, yet remains underutilized. The use of electronic media (e-media) decision aids improves saliency and fosters informed decision-making. This systematic review aimed to determine the effectiveness of CRC-screening promotion, using e-media decision aids in primary healthcare (PHC) settings. Three databases (MEDLINE, Web of Science, and the Cochrane Library) were searched for eligible studies. Studies that evaluated e-media decision aids compared to usual care or other conditions were selected. Quality was assessed by using Cochrane tools. Their effectiveness was measured by CRC-screening completion rates, and meta-analysis was conducted to calculate the pooled estimates. Ten studies involving 9393 patients were included in this review. Follow-up durations spanned 3–24 months. The two types of decision-aid interventions used were videos and interactive multimedia programs, with durations of 6–15 min. Data from nine feasible studies with low or some risk of bias were synthesized for meta-analysis. A random-effects model revealed that CRC-screening promotion using e-media decision aids were almost twice as likely to have screening completion than their comparisons (OR 1.62, 95% CI: 1.03–2.62, *p* < 0.05). CRC-screening promotion through e-media has great potential for increasing screening participation in PHC settings. Thus, its development should be prioritized, and it should be integrated into existing programs.

## 1. Introduction

Colorectal cancer (CRC) remains the third most common malignancy worldwide, and it was ranked the second leading cause of cancer deaths in 2018 [1]. Recent data from the World Health Organization confirm that CRC is the third most common cancer in men, after lung and prostate cancer, and the second most common cancer in women, following breast cancer [2]. The incidence of CRC has continued to increase in countries across various geographical areas and development levels, and in younger populations [3]. The constellation of factors associated with westernization, such as obesity, physical inactivity, high consumption of red meat, excessive alcohol intake, and smoking, have tremendously contributed towards the high CRC incidence in economically transitioning countries [4]. Further, the economic burden of CRC is substantial and is likely to increase over time, owing to its rising incidence trend, with most patients being diagnosed at late stages [5,6,7].

Screening average-risk adults (age 50 to 75 years) for CRC contributes to reduced mortality [8,9]. Nonetheless, public awareness and the participation rates of CRC screening in most countries remain low [9,10,11]. In countries where CRC screening is offered in primary healthcare (PHC) services, not many participants receive physician recommendations for the screening [12,13,14].

It is therefore essential to invest in health-promotion programs that target the general population, focusing on increasing knowledge of the benefits of CRC screening and promoting healthy lifestyles as a matter of disease prevention. Moreover, as many patients with CRC have lower education levels [15,16], it is necessary to develop low-level-literacy health-promotion decision aids that can capture their attention and comprehension. In this digital era, rapid and innovative advances in electronic gadgets and internet communication have turned electronic media (e-media) into a powerful health-promotion tool with great opportunities for health behavior modification, as it is relatively more attractive, inexpensive, and more sustainable [17,18,19,20]. E-media decision aids that provide and support audiovisual programs should integrate a hybrid approach for digital video that enables “online” and “offline” participation to support more inclusive health and well-being promotion, using digital technology [21]. Such decision aids can deliver informational materials in various formats, such as e-reminders by health professionals via email or text message, tele-counselling, videos, or interactive multimedia programs. Although it is contended that many digitized health-promotion strategies focus on peoples’ responsibility for their own health, some fail to recognize the social, cultural, and political dimensions of digital technology use [22]. Hence, such e-media should be developed according to existing cultural values.

This systematic literature review was aimed at assessing the effectiveness of e-media decision-aid interventions used in PHC settings, from identifying to reminding patients who have not responded to CRC screening. We considered a systematic literature review the most appropriate form of review to address the research question: how effective is e-media decision aid intervention, compared with a baseline or control group, in improving CRC screening among eligible patients in PHC settings? Thus, this review helps to identify future practice and strategies for improving CRC-screening effectiveness in PHC services.

## 2. Materials and Methods

### 2.1. Protocol Registration

The protocol for this systematic review was registered at Prospero (registration ID CRD42020220301) prior to the commencement of research to avoid duplication and minimize reporting bias.

### 2.2. Database Used

We searched the MEDLINE, Web of Science, and the Cochrane Library databases, because they were expected to contain relevant studies with numerous open-access full-text articles available online. The scope of MEDLINE is broadly defined and encompasses the areas used by health professionals engaged in basic research and clinical care, public health, health policy development, or related educational activities [23]. Web of Science contains a remarkable treasure trove of data on scientific content on a global scale that has been an indispensable resource for studying of science, technology, and knowledge [24]. The Cochrane Library comprises a collection of databases that contain different types of high-quality independent evidence to inform healthcare decision-making, and concentrated reports of randomized and quasi-randomized controlled trials [25].

### 2.3. Systematic Review Process

The literature search was performed from November 2020 to December 2020, and we searched for articles published from January 2010 up to 4 December 2020. The 10-year span was selected based on the integration of e-media and the Internet of Things, an important technical domain that has grown rapidly in the past decade and has received wide attention globally from a plethora of disciplines, including medical research [26]. The articles were selected in accordance with the Preferred Reporting Items for Systematic Reviews and Meta-Analyses or PRISMA guideline [27] (Figure 1).

### 2.4. Identification

A preliminary search was conducted to identify the appropriate keywords and to determine whether this review was feasible. The keywords were verified and validated by two public health physicians at the Universiti Kebangsaan Malaysia. The medical subject heading (MeSH) keywords used were as follows: (“colorectal cancer” OR “colon cancer” OR “bowel cancer” OR “rectal cancer” OR “colorectal neoplasm” OR “colorectal carcinoma”) AND (“screening” OR “detection” OR “prevention” OR “colonoscopy” OR “fecal occult blood” OR “FOBT”) AND (“media” OR “mass media” OR “electronic media” OR “video”) (Appendix A).

Potential additional studies were identified through reference tracking of systematic reviews during the database search. Given the extensive nature of the three databases, we did not search unpublished reports.

### 2.5. Screening

The initial search retrieved 964 articles (Medline = 124, Web of Science = 226, the Cochrane Library = 203, and reference tracking = 8). The search results were imported into the EndNote reference manager, and duplicates were removed (*n* = 312). Basic information from the remaining articles was exported to a Microsoft Excel sheet. Two authors (NSR and MII) independently reviewed all titles, abstracts, and references generated by the original search to identify articles for potential inclusion. A total of 644 titles and abstracts were screened based on their relevance to the inclusion and exclusion criteria, resulting in the exclusion of 572 articles, leaving 72 articles to be assessed for eligibility.

### 2.6. Eligibility

The inclusion criteria were limited to randomized or quasi-experiment trials of CRC-screening promotion interventions using e-media, based on the PICO framework: (a) population—eligible patients in PHC settings; (b) intervention—CRC-screening promotional tools using e-media; (c) comparison—usual care or otherwise specified; and (d) outcome—CRC-screening completion, either fecal occult blood test (FOBT) or colonoscopy or sigmoidoscopy uptake. In addition, eligible articles had to be published in English, with open-access full-text articles available online. We used the modified Australian National Bowel Cancer Screening Program (NBCSP) Quality Framework [28] as a guideline for identifying eligible patients and for optimizing their participation at PHC centers in selected studies (Figure 2). This framework was chosen because it illustrates a similar concept of the CRC-screening clinical pathway used in other countries. Articles were excluded if they failed to meet the inclusion criteria, or assessed interventions that involved surveillance colonoscopy or follow-up after cancer treatment.

A total of 57 articles did not meet the eligibility criteria. In the next stage, two authors (NSR and AMN) assessed the full articles independently and compared the results. To increase the reliability of the study selection, all differences were reconciled by consensus. In the full assessment of the remaining 15 articles, five were excluded for falling outside the scope of this review, resulting in the selection of a final 10 articles.

### 2.7. Data Extraction

At the final stage, NSR extracted data by using a standardized Excel spreadsheet, which was revised by another three authors (AMN, MRH, and MRM). The data extracted for mapping and analysis included author, year of publication, study country, study design, participants, sample size, type of intervention including content and duration, comparison, main outcome measures, and results. The outcome measures and operational definitions used for describing CRC-screening promotion effectiveness are defined in Table 1. The effectiveness of the primary outcome of interest (CRC-screening completion) was categorized into either *effective*, i.e., higher CRC-screening completion rate with statistical significance; *null*, i.e., higher or lower CRC-screening completion rate without statistical significance; or *not effective*, i.e., lower CRC-screening completion rate with statistical significance, and was reported as our final outcome summary.

### 2.8. Risk of Bias Assessment

We assessed study quality by using the Cochrane risk-of-bias tool for randomized trials (RoB 2) and the risk-of-bias tool for cluster randomized trials (RoB 2 CRT) because they are the most commonly used tool for randomized trials [29]. Bias was assessed in five distinct domains, where answers were required for one or more signaling questions. These answers led to evaluations of “low risk of bias”, “some concerns”, or “high risk of bias”. The evaluations within each domain led to an overall risk of bias evaluations for the result being assessed, which should enable the stratification of meta-analyses according to the risk of bias.

### 2.9. Data Analysis

All statistical analyses were performed by using Review Manager 5.4.1 (Cochrane, London, UK) [30]. We included “low risk of bias” and “some concerns” studies in the meta-analysis. Comparable data from studies were pooled by using forest plots. The odd ratios (ORs) with 95% confidence intervals (CIs) for dichotomous data were used as the effect measure and were reported as the primary outcome. Inter-study heterogeneity was assessed by using the *I*^2^ statistic for each pooled estimate. We used a random-effects model for heterogeneity (*p* < 0.05). Due to the possibility of clinical homogeneity, we performed subgroup analysis on the type of e-media decision aid used and the target population. The robustness of the results was evaluated with sensitivity analysis by excluding individual studies from each forest plot. Publication bias was assessed by using funnel plots.

## 3. Results

### 3.1. Characteristic of Studies

A total of 10 studies were included in this review. Eight studies were conducted in the United States [31,32,33,34,35,36,37,38], while the remaining two took place in New Zealand [39] and Iran [40]. Eight studies were randomized controlled trials [31,32,33,35,37,38,39,40]; another two had quasi-experimental designs [34,36].

Four studies followed their patients for a minimum of 3 months [32,35,38,39]; one for 4 months [40]; two for 6 months [33,37]; and one each for 12 months [36], 14 months [31], and the maximum duration of 24 months [34]. The majority of studies reported interventions targeting eligible patients [31,32,33,34,35,36,37,38,40]; one study targeted non-adherence patients [39]. More than half of the included studies focused on improving the screening participation of specific vulnerable populations that tend to be under-screened or never screened [32,33,34,35,36,38,39].

In total, 9393 patients were had follow-ups. The sample sizes ranged from 65 [38] to 5271 [39], with most studies taking place in a multi-clinic setting [31,32,33,34,35,36,40]. Table 2 provides an overview of the characteristics of the included studies.

### 3.2. E-Media Decision Aid Intervention Type, Content and Duration Length

The 10 studies yielded two types of decision aid interventions used for screening participation: videos, either with or without combination with other types of media tools [32,33,34,35,36,38,39,40], and interactive multimedia programs [31,37]. All contents pertained to the promotion of CRC screening and provided a basic overview of the disease and its screening options, purposely developed for easy understanding. Of the 10 included studies, four did not mention the intervention duration length; three articles reported the intervention duration length, which ranged from 6 min [39] to 15 min [33,40].

### 3.3. Effectiveness of CRC-Screening Promotion Using E-Media

#### 3.3.1. Studies Included in Qualitative Synthesis

CRC-screening completion (or participation or adherence to either FOBT or colonoscopy or sigmoidoscopy uptake) was used as a proxy for determining the effectiveness of CRC-screening promotion intervention using e-media. The final outcome summary derived from all studies was categorized accordingly (Table 3). Almost half of these studies [33,36,38,40] reported effective results from their significant statistical values (Figure 3). Half of them were deemed null due to their non-significant results, despite some studies yielded higher CRC-screening rates [31,32,35,37]. Interestingly, the statistical analysis of one study demonstrated not effective results, showing a lower CRC-screening completion rate in the intervention group [39].

#### 3.3.2. Studies Included in Quantitative Synthesis

Of the 10 studies included in this review, one [40] was not feasible for meta-analysis due to the lack of data. A total of 9145 samples were pooled from the nine feasible studies with either low or some risk of bias (Appendix A). The random-effects model revealed that there was a statistically significant positive effect of CRC-screening promotion using e-media interventions, where participants in the intervention groups were almost twice as likely to have CRC-screening completion (OR = 1.62, 95% CI: 1.03–2.62, *p* < 0.05) (Figure 4). Nonetheless, the clinical heterogeneity was high (*I*^2^ = 93%, *p* < 0.05), and the funnel plot generated was asymmetrical (Figure 5). The sensitivity analysis results were no different after each trial had been excluded.

In the subgroup analysis, two studies that used interactive programs as their e-media decision aid [31,37] reported a positive estimated pooled effect, wherein those in the intervention groups were more likely to complete their CRC screening (OR = 1.19, 95% CI 0.83–1.69, *p* > 0.05); however, it did not reach statistical significance (Figure 6). On the other hand, the estimated pooled effect targeting the Latino population retrieved from two studies [32,38] showed a significant positive effect of e-media interventions for CRC-screening promotion. In those studies, participants in the intervention groups were four times more likely to complete their screening compared to those in the comparison groups (OR = 4.06, 95% CI 4.63–10.09, *p* < 0.05) (Figure 7). The analyses of heterogeneity in these two subgroups were able to prove homogeneous results (*I^2^* = 0%, *p* < 0.05).

#### 3.3.3. Secondary Outcome

The other main outcomes (Table 1) from the CRC-screening promotion interventions being reported are presented in Table 2, numbering a total of five secondary outcomes included in this review. Bartholomew et al. [39] reported evidence of a lower rate of spoiled FOBT kit return in the intervention groups than in the control groups in their two study populations (12.4% vs. 33.1% in Māori and 21.9% vs. 42.1% in a Pacific population). However, it did not reach statistical significance. Gwede et al. [32] found a significant association between CRC-screening promotion intervention with greater increases in CRC awareness and susceptibility, while cancer worry increased significantly in their comparison group. The ability to state CRC-screening test preference, readiness to receive CRC screening, and requesting CRC-screening tests were among the other main outcomes reported by Miller et al. [37], who found that intervention participants had significantly higher rates of CRC-screening preference (84% vs. 55%) and readiness to receive screening (52% vs. 20%) compared to their control counterparts. The rate of CRC-screening tests requested was also higher (30% vs. 21%); however, this was not significant. In general, these five secondary outcomes show the positive effects of CRC-screening interventions, regardless of their statistical significance values.

## 4. Discussion

This systematic literature review highlights the effectiveness of CRC-screening promotion practiced in PHC settings and research opportunities for improving CRC-screening participation for eligible and non-adherent patients in bowel cancer screening programs. This review makes a distinction between the types of e-media decision aid used for promoting CRC screening to optimize recruitment participation and follow-up, and towards better integration of CRC screening into existing PHC prevention pathways.

Our findings demonstrate that the rates of CRC-screening completion were higher among those who received e-media decision aid interventions in the majority of studies (8/10), of which four reached statistical significance (Table 3). From the meta-analysis, it is evident that videos and interactive multimedia programs can significantly prompt previously unscreened people to choose their preferred screening test, thus resulting in higher rates of CRC-screening completion. This result also suggests that public population-based CRC-screening intervention programs have been implemented heterogeneously across countries and regions, with significant CRC-screening completion rates seen in vulnerable Latino populations. The high clinical heterogeneity (*I*^2^ = 93%, *p* < 0.05) shown may possibly reflects the variety of the population groups, sample sizes and decision aid contents including duration. In addition, the asymmetrical shape of funnel plot suggests for the existence of publication bias. This type of bias is induced by the fact that research with statistically significant results is more likely to be submitted and published than work with null or non-significant results. Thus, it poses a threat to the validity of such analyses. The implication of having publication bias may lead to an incorrect, usually over optimistic conclusion. Therefore, cautious interpretation is vital upon its existence. Finally, all secondary outcomes showed positive effects of CRC-screening promotion that uses e-media interventions when compared with their respective controls. Therefore, we conclude that e-media decision-aid interventions are effective for promoting CRC screening.

The effectiveness of health interventions has been linked to the use of health behavioral science theories. The health-belief model (HBM) has been applied most often for health concerns that are prevention-related and asymptomatic, such as early cancer detection, where beliefs are equally or more important than overt symptoms [41]. The HBM theorizes that people’s beliefs about whether or not they are at risk for a disease, and their perceptions of the benefits of taking action to avoid it, influence their readiness to take action within the core construct of *perceived susceptibility*, *perceived severity, perceived benefits, perceived barriers, cues to action,* and *self-efficacy* [42]. These concepts have been explicitly applied in studies that showed effective results, where the videos and interactive multimedia programs used were able to inform participants of their *perceived susceptibility* by conveying the importance of screening, including CRC risk factors [33,34,38,40], persuade them to overcome *perceived barriers* [36], provide of *cues to action* by addressing the need to change their screening behavior [31] and prompt them towards *self-efficacy* to complete CRC screening [32,37].

While the HBM focuses on the individual, it also recognizes and addresses the social context in which health behaviours take place [43]. Thus, the infusion of the culturally tailored decision-aid concept also helps to enhance the positive effect of health interventions, as shown by some studies [32,34,38] in the present review. This result is also congruent with the findings of other studies, where viewing culturally tailored decision aids can significantly increase patients’ knowledge of CRC-screening recommendations and options, with a significant reduction in their decisional conflict and improved self-advocacy [44,45].

Through e-media platforms, not only can decision aids such as health-promotion videos exert great influence in patient decision-making, but they can also empower the willingness of many adults to use e-media to promote cancer screening to their peers [46]. Therefore, the studies in the present review should have adopted the Technology Acceptance Model [47] theory, along with the HBM, which highlights the central role of *perceived usefulness, perceived ease-of-use,* and *users’*
*acceptance* constructs towards e-media or internet use [48]. Nonetheless, none of the included studies mentioned the theory or others of its kind. The constructs of such a theory are important, as health experts have concluded that there is a need for more comprehensive videos that are easily identifiable by patients [49]. This probably explains why we did not find many *effective-cum-statistically significant* interventions in this review.

Apart from the *effective* studies discussed above, two studies [35,39] reported lower rates of CRC-screening completion in their intervention groups. When comparing with other studies that had eligible patients as their study population, Bartholomew and colleagues [39] examined a population of those non-adherent to initial screening invitation. Knowing at the beginning of the study that the target population was non-adherent, the researchers should have applied all of the core constructs of the HBM theory, which the study lacked. In the second study that resulted in lower CRC-screening completion rate, Larkey and colleagues [35] used a video drama of “Papa” receiving CRC screening as the intervention. Reflecting on that study, there was a possible lack of the *self-efficacy* construct at the end of the story, and the participants merely perceived the video as a form of entertainment. Hence, decision aids that only hint at why screening interventions are given are not effective in comparison with the studies that were effective.

The literature search showed that the low uptake in CRC-screening reported in many parts of the world were more prominent among underserved populations such as the uninsured, recent immigrants, and in the most ethnically diverse areas, with a striking gradient according to socioeconomic status [50,51]. This probably explains why more studies on CRC-screening interventions in the present review focused on certain target populations (Table 2). Besides that, a majority of the studies in this review were conducted in the United States, hence possibly indicating its higher CRC burden than that in other countries and that its health authorities have taken the necessary health-promotion activities and research action.

E-media decision aids were often accompanied by other types of decision aids, such as patient navigator, brochure, pamphlet, or reminder letter, as shown in studies in the present review. In some other studies, multilayer-screening interventions involving patient navigators, such as nurses and other community health workers, were also used [52,53,54,55]. Of all the types of e-media decision-aids, videos were more frequently used compared with the other types, such as multimedia interactive programs or mobile phone reminders. The duration of decision aids used was 6–15 min, and was thought to be quite lengthy [56]; however, there was a significant association between video length and level of usefulness found [57].

Although we have commented on the positive effect, there were also some no-effect findings on the rate of CRC-screening completion among patients in PHC following video-type intervention. Zapka and colleagues discuss such results explicitly, where their studied samples were primarily middle-class white people who had high screening rates at baseline, and the trial was conducted during a period of increased health insurance coverage for lower endoscopy procedures with wide public media attention on CRC screening [58]. Similarly, studies conducted in groups with suboptimal CRC-screening rates reported no significant differences in the participants’ attitudes, norms, or intentions regarding CRC-screening uptake [44,59].

These findings suggest that future theory-guided trial interventions are needed, as well as the need to examine health behavior moderators and mediators. Issues that are in line with the HBM theory that could explain negative health behavior may also be considered in developing CRC-screening decision aids [60]; for example, the narration of personal experiences of cancer survivors could provide many *cues to action* for those who have never encountered this potentially deadly but preventable disease, and thus should be incorporated into health promotion activities.

### Limitation

Reporting and publication bias may have affected our findings. The independent effect of multicomponent elements of e-media decision-aid interventions was often uncertain, as can be inferred from the high heterogeneity in meta-analysis. Despite the limitations, the findings are relevant to settings in private practice or through an organized screening program, given the role of PHC services for preventive care follow-up. To our knowledge, this is the first review that focuses on e-media decision aids in the CRC-screening pathway.

## 5. Conclusions

Overall, this review provides effective evidence of intervention studies that used e-media platforms for promoting CRC screening. In other words, e-media decision aids have been proven to have great potential for increasing CRC-screening participation in the era of rapid electronic gadget revolution in combination with the Internet of Things. If integrated appropriately in PHC settings, these interventions could act as an effective learning interface for patients with CRC and their families. Therefore, development and application of culturally tailored e-media decision aids aimed at increasing CRC-screening completion should be prioritized. However, healthcare providers and organizations must be aware of the limitations and pitfalls of these platforms and must address them appropriately.

## Figures and Tables

**Figure 1 ijerph-18-08190-f001:**
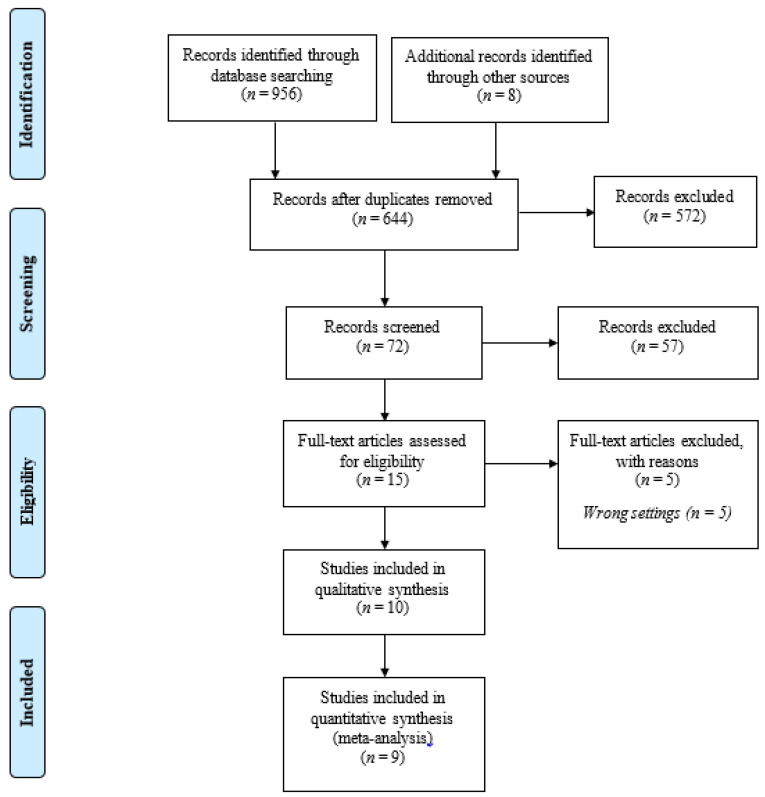
Study selection according to PRISMA guideline.

**Figure 2 ijerph-18-08190-f002:**
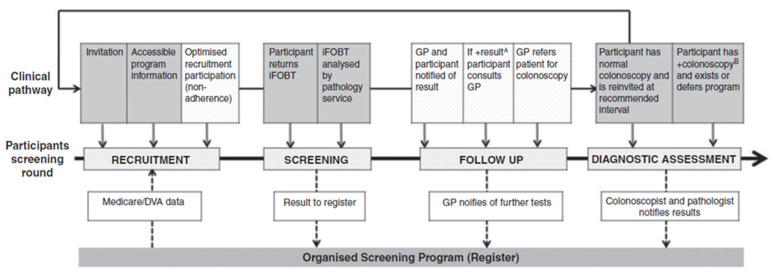
The modified National Bowel Cancer Screening Program Quality Framework, version 2. ^A^ A positive (+) result means that blood was detected in the completed immunochemical fecal occult blood test (iFOBT). ^B^ A positive colonoscopy is identified by reporting one of the following: tubular adenoma, tubulovillous adenoma, villous adenoma, sessile serrated adenoma, traditional serrated adenoma, adenoma not otherwise classified, or carcinoma.

**Figure 3 ijerph-18-08190-f003:**
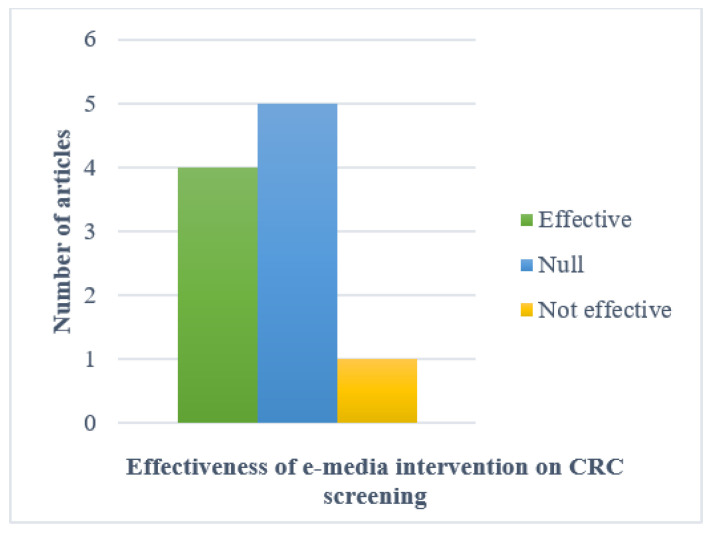
Comparison of effectiveness of e-media intervention on CRC screening (*N* = 10).

**Figure 4 ijerph-18-08190-f004:**
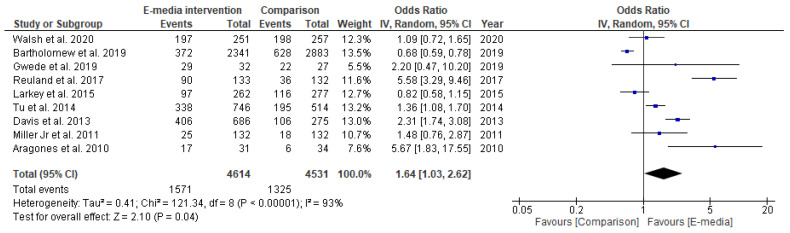
Random-effects forest plot for studies eligible for meta-analysis (*n* = 9). Box size represents study weighting. Diamond represents overall effect size and 95% CI.

**Figure 5 ijerph-18-08190-f005:**
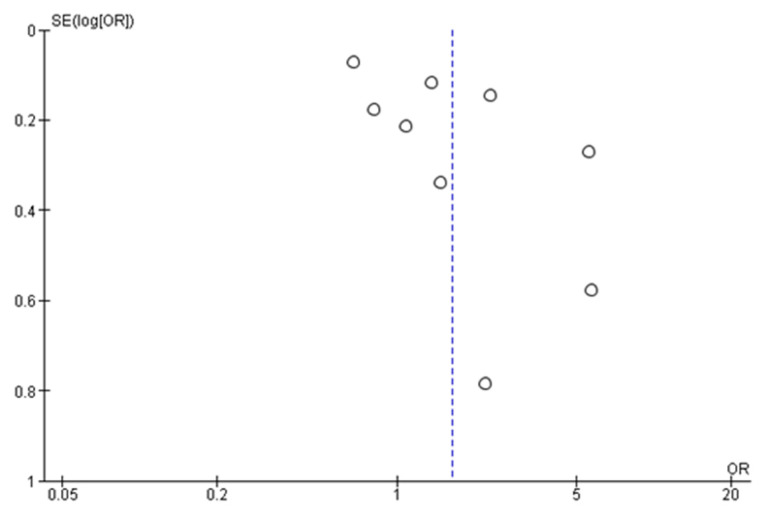
Funnel plot for studies eligible for meta-analysis (*n* = 9).

**Figure 6 ijerph-18-08190-f006:**
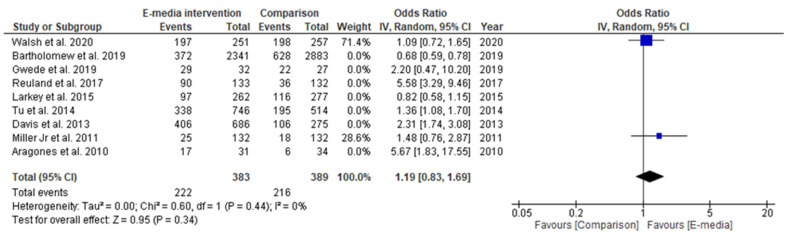
Random-effects forest plot for studies using interactive programs as intervention (*n* = 2). Box size represents study weighting. Diamond represents overall effect size and 95% CI.

**Figure 7 ijerph-18-08190-f007:**
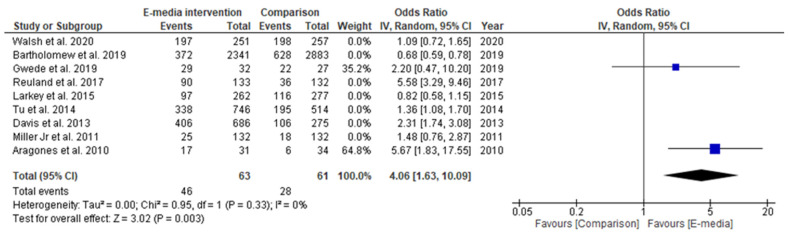
Random-effects forest plot for studies targeting Latino populations (*n* = 2). Box size represents study weighting. Diamond represents overall effect size and 95% CI.

**Table 1 ijerph-18-08190-t001:** Outcome measures and operational definitions used.

Outcome Measures		Operational Definition
Primary	CRC-screening completion rate	Completion rate or participation rate or adherence rate to either FOBT or colonoscopy or sigmoidoscopy uptake.
Secondary	Spoiled kit return rate	Spoiled FOBT kits are samples which cannot be analyzed for one reason or another (e.g., no date on the specimen).
	CRC-screening awareness and belief	Awareness score was calculated by summing the points earned for all awareness items. A scale assessing influence of religious beliefs on medical-decision-making, such as CRC screening, was used.
	Ability to state CRC-screening test preference	Assessment on the post-program survey by asking patients which CRC-screening test they would want if all tests were free.
	Readiness to receive CRC screening	Determined by comparing patients’ readiness stage after the intervention program to their baseline stage.
	Ordered CRC-screening tests	Determined by patients’ chart review.

Abbreviation: CRC = colorectal cancer, FOBT = fecal occult blood test.

**Table 2 ijerph-18-08190-t002:** Table of evidence from all ten studies in this review (*N* = 10).

No.	Author (Year)	Country	Study Design (Follow-Up)	Participants (Sample Size)	Type of e-Media Decision-Aid Used	E-media Decision-Aid Content (Duration)	Comparison	Main Outcome Measures	Results
1	Walsh et al. (2020)	San Francisco, United States	RCT (14 months)	Patients with upcoming follow-up or preventive appointments (508, 6 clinics)	Interactive multimedia program (“Video Doctor” and “Provider Alert” paper, named “PreView” (Preventive Video Education in Waiting Areas))	Series of questions (health assessment, prior cancer screening, and readiness to change cancer screening behavior) Stage of change assessment “Video Doctor” conversation between a patient and a physician Participants’ receipt of “Provider Alert” individualized messages (Not mentioned)	Video about healthy lifestyle	^c^ CRC-screening completion ^f^	CRC-screening completion rate was higher in the intervention group than in the control group (78.5% vs. 77.0%) **.
2	Bartholomew et al. (2019)	Auckland, New Zealand	RCT (3 months)	Māori and Pacific residents, non-adherent to initial screening invitation (5271, clinic numbers not mentioned)	Video (DVD and a reminder letter)	The importance of CRC screening The ease of the test The nature of return of results Participants’ positive experiences of the diagnostic follow-up test (6 min)	^d^ Usual reminder letter	^a^ CRC-screening participation ^e^ Spoiled FOBT kit return	CRC-screening participation rate was lower in the intervention groups than in the control groups, for both ethnic groups (13.6% vs. 25.9% in Māori and 10.1% vs. 18.4% in Pacific) *. Spoiled kit rates were lower in the intervention groups than in the control groups (12.4% vs. 33.1% in Māori and 21.9% vs. 42.1% in Pacific) **.
3	Gwede et al. (2019)	Florida, United States	RCT (3 months)	Latinos population (76, 2 clinics)	Video (DVD with fotonovela booklet and FIT kit, called “LCARES” (Latinos CRC Awareness, Research, Education, and Screening))	Constructs of a preventive health model for CRC screening, e.g., salience and coherence, cancer worry and self-efficacy Culturally tailored (Not mentioned)	Booklet about CRC-screening promotion in Spanish and a FIT kit	^a^ CRC-screening completion CRC screening awareness and beliefs	CRC-screening completion rate was higher in the intervention group than in the control group (90% vs. 83%) **. The intervention group was associated with greater increases in CRC * awareness and * susceptibility, * cancer worry increased more in the comparison group.
4	Besharati et al. (2017)	Hamadan City, Iran	Cluster RCT (4 months)	Iranian adults (248, 8 clinics)	Video ((educational package consists of video, discussion, role play, a reminder pack of postcards, and pamphlet). Group 1: Educational package and free FOBT; Group 2: only Educational package; Group 3: free FOBT)	Educational video entitled “Being a winner in life: How to prevent CRC” Discussion on perceived susceptibility, social support, barriers, benefits and intention (15 min)	Usual care given with survey about determinants of CRC-screening behavior	^a^ CRC-screening completion	CRC-screening completion rates were higher in the three intervention groups than in the control group (87.1%, 61.3%, and 54.8 in Group 1, Group 2, and Group 3, respectively, vs. 1.6%) *.
5	Reuland et al. (2017)	North Carolina and New Mexico, United States	RCT (6 months)	Vulnerable low income population (265, 2 clinics)	Video (decision aid and patient navigation (employees of the clinic or its affiliated health system))	Importance of CRC screening Review screening test options Selection of a brochure of their CRC-screening readiness (15 min)	Video on food safety and ^d^ usual care	^b^ CRC-screening completion	CRC-screening completion rate was higher in intervention group than in the control group (68% vs. 27%) *.
6	Larkey et al. (2015)	Arizona, United States	RCT (3 months)	Low-income patients (545, 15 clinics)	Video	A drama about “Papa” receiving CRC screening Information on risk Reflecting elements for behavior change in cultural elements Creating dramatic tension— what will Papa’s test results be? (7 min)	Usual care given with instrument estimating level of personal cancer risk based on the Harvard Cancer Risk Index	^c^ CRC-screening adherence	CRC-screening adherence rate was lower in the intervention group than in the control group (37% vs. 42%) **.
7	Tu et al. (2014)	Washington, United States	Quasi experiment (24 months)	Vietnamese immigrants (1260, 2 clinics)	Video ((DVD) and a pamphlet)	Motivational education promoting on CRC screening Culturally tailored (Not mentioned)	Usual care (FOBT ordered by primary care providers)	^c^ CRC-screening adherence	CRC-screening adherence was higher in the intervention group than in the control clinic (45% vs. 38%, ^g^ AOR 1.42; 95% CI 0.95–2.15) **. Among those who were non-adherent at baseline, overall CRC-screening adherence was higher in the intervention group than in the control group (47.3% vs. 34.5%, ^g^ AOR 1.70; 95% CI 1.05–2.75) *.
8	Davis et al. (2013)	Louisiana, United States	Quasi experiment (12 months)	Low-income, uninsured patients in predominantly rural areas (961, 8 clinics)	Video (Educational strategy (enhanced usual care, pamphlet, video, and FOBT instructions). Group 1: Educational strategy; Group 2: Educational strategy and nurse support to encourage CRC-screening completion)	Patients discussing on barriers and facilitators to screening and a physician making a recommendation while showing key steps in FOBT completion (Not mentioned)	Enhanced usual care (recommendation for CRC screening and FOBT kit)	^a^ CRC-screening completion	CRC-screening completion rate was higher in the intervention groups than in the control group (57.1% in education arm and 60.6% in nurse support arm vs. 38.6% of control arm) *. Those in the nurse support arm were more likely to be screened than those in the control arm (^h^ AOR 1.6; 95% CI 1.06–2.42) *, but no more likely to be screened than those in the educational arm (^h^ AOR 1.18; 95% CI 0.97–1.42) **. Those in the educational arm were not more likely to be screened than those in control arm (^i^ AOR 1.36; 95% CI 0.85–2.18) **.
9	Miller et al. (2011)	North Carolina, United States	RCT (6 months)	Patients scheduled for routine visits (264, 1 clinic)	Interactive multimedia web-based program (named “CHOICE” (Communicating Health Options through Interactive Computer Education, version 6.0 W))	Overview of CRC screening Education options (to learn about a specifıc test, view tests comparison, or end the program) Choice of screening decision A corresponding handout Encouragement for screening decision discussion with healthcare providers (6.3 min overview)	Interactive web-based program about prescription drug refılls and safety	^c^ CRC-screening completion Ability to state screening test preference, readiness to receive screening, and ordered screening tests	CRC-screening completion rate was higher in the intervention group than in the control group (19% vs. 14%, ^4^AOR 1.7; 95% CI 0.88–3.2) **. The rates of ability to state screening test preference, readiness to receive screening, and ordered screening tests were all higher in the intervention group than in the control group (84% vs. 55% *, 52% vs. 20% *, 30% vs. 21%, ^j^ AOR 1.6; 95% CI 0.97–2.8) **.
10	Aragones et al. (2010)	New York, United States	RCT (3 months)	Latino immigrants (65, 1 clinic)	Video ((and a brochure for patient) with a paper-based reminder for their physicians)	Education about CRC-screening modalities, prevention, and risk factors Culturally tailored (11 min)	^d^ Usual care	^c^ CRC-screening completion	CRC-screening completion rate was higher in the intervention group than in the control group (55% vs. 18%, ^k^ AOR 5.4; 95% CI 1.6–18.5) *.

Abbreviation: AOR = adjusted odd ratios, CI = confidence interval, CRC = colorectal cancer, FIT = fecal immunochemical test, iFOBT = immunohistochemical fecal occult blood test, FOBT = fecal occult blood test, RCT = randomized controlled trial. ^a^ FIT/FOBT/iFOBT uptake. ^b^ FIT/FOBT/iFOBT or colonoscopy uptake. ^c^ FIT/FOBT or colonoscopy or sigmoidoscopy uptake. ^d^ No further description. ^e^ Spoiled FOBT kits are samples which cannot be analyzed for one reason or another (e.g., no date on the specimen). ^f^ Measure completion of other cancer (breast, cervical) screenings, and prostate screening discussion. ^g^ Adjusted for age, gender, insurance status, primary care provider category, language concordance with primary care provider, and continuity index. ^h^ Adjusted for age, race (African American vs. Caucasian and Hispanic), sex, and literacy (2 categories). ^i^ Adjusted for age and race (African American vs. Caucasian and Hispanic). ^j^ Adjusted for marital status, health insurance, literacy level, baseline readiness stage, and provider training level. ^k^ Adjusted for patient variables (age, gender, education level, insurance status, acculturation level, and English proficiency) and physician variables (attending vs. resident, Spanish fluency). * *p* < 0.05, ** *p* > 0.05.

**Table 3 ijerph-18-08190-t003:** Outcome summary for CRC-screening completion rates using e-media interventions (*N* = 10).

Study (Year)	Outcome (Intervention vs. Control)				Outcome Summary
	Higher CRC-Screening Completion Rate with Statistical Significance	Higher CRC-Screening Completion Rate without Statistical Significance	Lower CRC-Screening Completion rate with Statistical Significance	Lower CRC-Screening Completion Rate without Statistical Significance	
Walsh et al. (2020)		√			Null
Bartholomew et al. (2019)			√		Not effective
Gwede et al. (2019)		√			Null
Besharati et al. (2017)	√				Effective
Reuland et al. (2017)	√				Effective
Larkey et al. (2015)				√	Null
Tu et al. (2014)		√			Null
Davis et al. (2013)	√				Effective
Miller et al. (2011)		√			Null
Aragones et al. (2010)	√				Effective

## Data Availability

The data presented in this study are available in this article.

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
