# Peer review of "Effectiveness of Colorectal Cancer Screening Promotion Using E-Media Decision Aids: A Systematic Review and Meta-Analysis"

_ijerph, 2021, doi:10.3390/ijerph18158190_

Round 1
Reviewer 1 Report
The Authors performed a systematic literature review to assess the effectiveness of electronic media as decision-aid interventions to be used in primary health care settings, from identifying to reminding patients who have not responded to colorectal cancer screening.
The topic is relevant and the article is of interest. Methodology is robust and the overall the mansucript is scientifically sound. I have a minor comment:
- Line 124: (c) Comparison – usual care or otherwise specified; Can you please specify what do you mean by Usual Care in the C of the PICO criteria.
- What is usual care and how did you double-check that the comparator belongs to this category?
Author Response
Dear Reviewer 1,
Please see the attachment.

Reviewer 2 Report
In my opinion, the submitted manuscript deals with an interesting and a current topic that studies the effectiveness of searching for patients with the use of electronic media, and may be accepted after minor revision.
- I would appreciate if authors could add a short paragraph (in the Introduction section?) related to the risk factors of CRC, and CRC prevention. In my opinion, neither colonoscopy nor faecal occult blood tests are cancer prevention – both are relly early diagnosis, which increases the chances of successful cancer therapy. When reading the article one may get the impression that screening is a method of prevention and there are no others.
Authors can use the proposed publications or choose any other on this subject:
O'Keefe, S. Diet, microorganisms and their metabolites, and colon cancer. Nat Rev Gastroenterol Hepatol 13, 691–706 (2016). https://doi.org/10.1038/nrgastro.2016.165
Keum, N., Giovannucci, E. Global burden of colorectal cancer: emerging trends, risk factors and prevention strategies. Nat Rev Gastroenterol Hepatol 16, 713–732 (2019). https://doi.org/10.1038/s41575-019-0189-8
Gravitz, L. Chemoprevention: First line of defence. Nature 471, S5–S7 (2011). https://doi.org/10.1038/471S5a
Brenner, H., Chen, C. The colorectal cancer epidemic: challenges and opportunities for primary, secondary and tertiary prevention. Br J Cancer 119, 785–792 (2018). https://doi.org/10.1038/s41416-018-0264-x
Maniewska, J., Jeżewska, D. Non-Steroidal Anti-Inflammatory Drugs in Colorectal Cancer Chemoprevention. Cancers, 13(4), 594 (2021) https://doi.org/10.3390/cancers13040594
- Authors explained that they considered open-acess articles only (line 127). In my opinion this information should also be added to the section 2.2, where they explain, that they have searched Medline, Web of Science, and Cochrane Library databases. It should be presicesly written, if they serched these databases for open-acess articles only.
- In Table 1 there is a dot (full stop) missing (in line „Spoilted FOBT kits…”). Authors can also use semicolons instead of dots in the tables, but should be consistent.
- The formatting of the Table 2 could be better. Maybe a horizontal table setting on the page, instead of a vertical one, would improve transparency?
- In section 3.3.3 there are references missing – Bartholomew ([38]?), Gwede ([31]?) and Miller Jr ([36]?).
Author Response
Dear Reviewer 2,
Please see the attachment.
